

# Identifying zooplankton community changes between shallow and upper-mesophotic reefs on the Mesoamerican Barrier Reef, Caribbean

Dominic A. Andradi-Brown[1,2], Catherine E. I. Head[1], Dan A. Exton[2], Christina L. Hunt[1,2], Alicia Hendrix[2], Erika Gress[2] and Alex D. Rogers[1]

[1] Department of Zoology, University of Oxford, Oxford, Oxfordshire, United Kingdom
[2] Operation Wallacea, Old Bolingbroke, Lincolnshire, United Kingdom

## ABSTRACT

Mesophotic coral ecosystems (MCEs, reefs 30–150 m) are understudied, yet the limited research conducted has been biased towards large sessile taxa, such as scleractinian corals and sponges, or mobile taxa such as fishes. Here we investigate zooplankton communities on shallow reefs and MCEs around Utila on the southern Mesoamerican Barrier Reef using planktonic light traps. Zooplankton samples were sorted into broad taxonomic groups. Our results indicate similar taxonomic zooplankton richness and overall biomass between shallow reefs and MCEs. However, the abundance of larger bodied (>2 mm) zooplanktonic groups, including decapod crab zoea, mysid shrimps and peracarid crustaceans, was higher on MCEs than shallow reefs. Our findings highlight the importance of considering zooplankton when identifying broader reef community shifts across the shallow reef to MCE depth gradient.

## INTRODUCTION

Tropical coral ecosystems are some of the most diverse ecosystems on the planet (*Sala & Knowlton, 2006*), with light-dependent scleractinian corals extending from the surface down to approximately 150–165 m depth (*Maragos & Jokiel, 1986*; *Kahng & Maragos, 2006*). Yet most coral reef research is heavily biased towards the shallows (<30 m) (*Menza, Kendall & Hile, 2008*) because of the difficulties associated with accessing deeper reefs. Mesophotic coral ecosystems (MCEs; *Hinderstein et al., 2010*), reefs from 30 m to approximately 150 m, are increasingly recognized as containing important components of reef diversity both as refuge habitats for shallow reefs and also as unique ecological assemblages that need protection in their own right (*Bongaerts et al., 2010*; *Bridge et al., 2013*; *Andradi-Brown et al., 2016a*).

Most of the limited work on MCEs has focused on large sessile taxa, such as scleractinian corals and sponges, or large mobile taxa such as fishes (*Kahng et al., 2010*; *Kahng, Copus & Wagner, 2014*). It has, however, been estimated that 168,000 invertebrate species have

Corresponding author
Dominic A. Andradi-Brown,
dandradibrown@gmail.com

been described on coral reefs (*Ruppert, Fox & Barnes, 2003*; *Stella et al., 2011*), far greater than the approximately 5,000 fish species and 700 scleractinian coral species currently recognized (*Veron, 2000*; *Bellwood, Renema & Rosen, 2012*). Most biodiversity on reefs is therefore comprised of small mobile invertebrates, many of which are cryptic and found associated with other sessile reef fauna or in the zooplankton (*Fautin et al., 2010*; *Plaisance et al., 2011*; *Head et al., 2015*).

Zooplankton is comprised of a diverse range of organisms with different components normally classified into coarse groups based on size, for example mesozooplankton range from 0.2 to 20 mm and macrozooplankton from 2 to 20 cm (*Johnson & Allen, 2012*). In the Caribbean, zooplankton recorded adjacent to coral reefs has generally been found to be dominated by copepods with amphipods, isopods, polychaetes, shrimp larvae and crab larvae all present but at lower abundances (*Heidelberg, Sebens & Purcell, 2004*; *Heidelberg et al., 2010*). Previous studies have suggested major factors affecting zooplankton abundance on reefs include currents, active zooplankton avoidance behaviour of reef habitats and differing exploitation patterns by predators (*Motro, Ayalon & Genin, 2005*; *Yahel, Yahel & Genin, 2005*; *Heidelberg et al., 2010*). Approximately 20% of reef invertebrates are crustaceans, making them one of the largest and most speciose groups on coral reefs (*Kramer, Bellwood & Bellwood, 2014*) and an important component of reef fish diets (*Randall, 1967*). Many zooplankton can exhibit active swimming behavior to avoid predation or areas with higher risk of predation (*Haury, Kenyon & Brooks, 1980*; *Ohman, 1988*). Planktivorous fish predation pressure is thought to lead to depleted plankton abundance over reefs, however, planktivorous fishes tend to feed <1.5 m over the reef and so this effect is limited to close to the reef surface (*Motro, Ayalon & Genin, 2005*).

On MCEs zooplankton is particularly important as a food source, because zooplanktivores are widely observed as the dominant fish trophic group globally, for example in the Red Sea (*Brokovich et al., 2008*), in Hawaii (*Pyle et al., 2016*) and in the Marshall Islands (*Thresher & Colin, 1986*). These patterns are also widely found in the Caribbean (*Garcia-Sais, 2010*; *Bejarano, Appeldoorn & Nemeth, 2014*; *Andradi-Brown et al., 2016b*) with a study suggesting that approximately 60% of MCE reef fish feed on zooplankton and mobile-invertebrates on the Mesoamerican Barrier Reef, Caribbean (*Andradi-Brown et al., 2016b*). Shallow-reef corals also feed on zooplankton, which provides an important additional energy source to that provided by their symbionts (*Ferrier-Pagès et al., 2003*). On MCEs corals are believed to increase heterotrophic feeding because of low light availability, likely making them more dependent on zooplankton than their shallow counterparts (*Fricke, Vareschi & Schlichter, 1987*; *Mass et al., 2007*; *Lesser et al., 2010*). In addition, invertebrate groups such as decapods have important functional roles in maintaining fish health. For example, cleaning fish of parasites, e.g., cleaner shrimp (*Becker & Grutter, 2004*), and defending coral colonies from predators and clearing excess sediment thus preventing smothering of coral polyps, e.g., *Trapezia* crabs (*McKeon & Moore, 2014*).

Despite the important roles zooplankton are likely to have on MCEs, few studies have documented MCE zooplankton communities and how they differ from those on shallow reefs. Here, we investigate the mesozooplankton and macrozooplankton community on

shallow reefs and upper-MCEs on the Mesoamerican Barrier Reef, Caribbean, to identify differences in abundance, biomass, and community structure across the depth gradient.

## METHODS

Surveys were conducted on the south shore of Utila, Bay Islands, Honduras. Utila is located off the north shore of Honduras, with its reefs forming the southern extent of the Mesoamerican Barrier Reef. Off the south shore of Utila, shallow reefs form a spur and groove system, with a reef slope down to approximately 35 m where the seabed flattens and a patch reef MCE is formed. From these MCE patch reefs the south shore seabed continues to gently slope to approximately 70–80 m before rising to the Honduran mainland. Three replicate light trap deployments were conducted at 15 m (shallow) and 40 m (MCE) at three sites: Coral View, Black Coral Wall and Little Bight (Fig. 1, see Table S1 for GPS locations) during July–September 2015. Light traps were built following *Jones (2006)*. Traps were modified from these specifications to use twelve 12 V light-emitting diodes (LEDs) powered by a 12 V, 4,800 mAh rechargeable lithium ion battery as the light source in each trap. LEDs were white light emitting with each LED having a luminous intensity of 12,000–14,000 mcd and wavelength of 5,000–6,500 nm. Light traps were deployed by divers 0.5 m above the reef at each depth during the afternoon. They were activated with a digital timer set to illuminate the trap 30 min before sunset and remain lit until sunrise the following morning. Divers recovered the light traps at 7:30 am the morning following deployment. Sites and depths were surveyed over multiple nights, with no more than two traps deployed at a site in a single night. All light traps were placed a minimum of 20 m distance from previous light trap deployments, and where two traps were placed at the same site on the same night these were separated by a minimum of 50 m. Research permits for this work were issued to Operation Wallacea by the Instituto de Conservación Forestal (ICF), Honduras, permit number: ICF-261-16. As the focus of the work was on invertebrates, and no higher vertebrates were involved, ethical review was not required.

Samples were sorted following groupings used in *Johnson & Allen (2012)* into broad taxonomic and developmental groups readily identifiable in the field with the use of a dissecting microscope. These groups were: (i) arrow worms, (ii) barnacle larvae, (iii) cladocerans, (iv) copepods, (v) decapod crab zoea, (vi) decapod shrimp zoea, (vii) decapod crab megalopae, (viii) lobster phyllosoma, (ix) mantis shrimp larvae, (x) mysid shrimps, (xi) peracarid crustaceans, (xii) oligochaetes, (xiii) polychaetes, (xiv) mites, (xv) urochordates and (xvi) fish larvae. All sampled individuals >2mm were counted to give abundance, and all individuals regardless of size were sorted and dry weighed to record biomass. Raw data is available in Data S1 and code in Data S2.

Nonmetric multidimensional scaling (NMDS) and permutational multivariate analysis of variance (PERMANOVA) were used to visualize and test for differences in abundance and biomass between the two depths based on Bray-Curtis dissimilarities on a fourth root transformed matrix (*Anderson, Gorley & Clarke, 2008*), while differences in richness were tested using a Euclidean PERMANOVA. Transformed data were used to reduce the influence of the most abundant taxonomic groups when assessing community differences
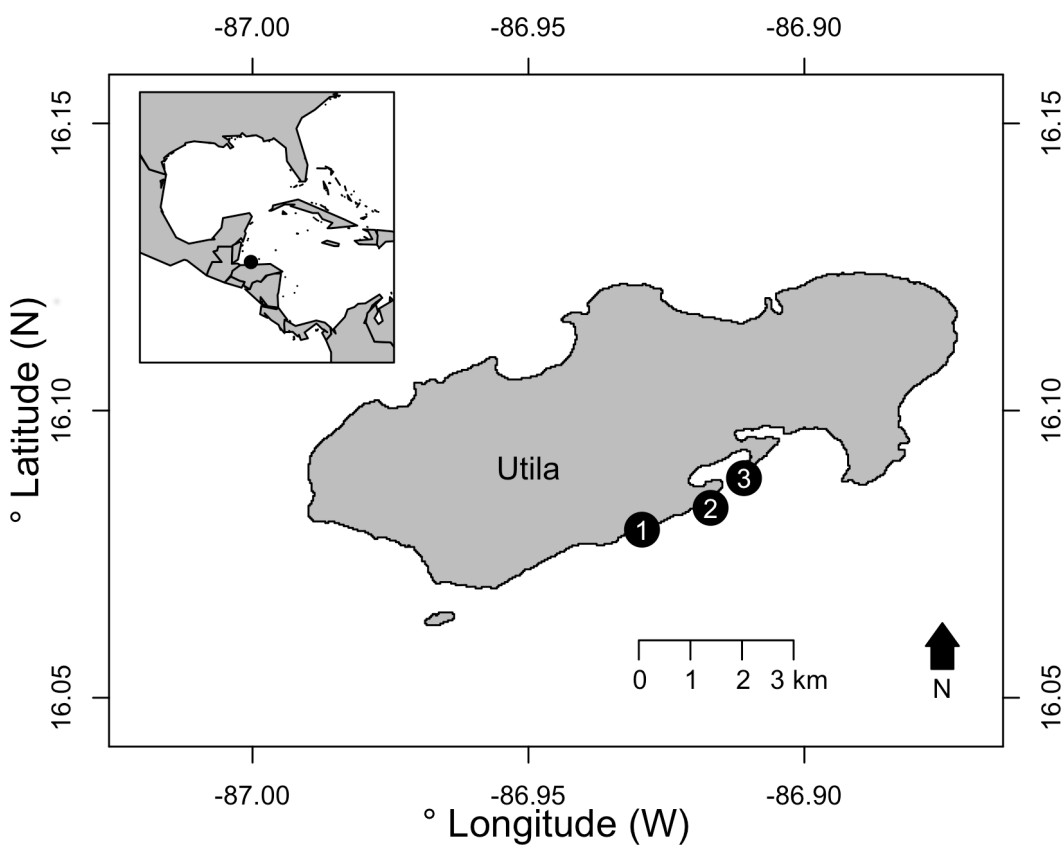

**Figure 1** **The three survey sites on the south shore of Utila, Bay Islands, Honduras.** Sites were: (1) Little Bight, (2) Black Coral Wall and (3) Coral View. See Table S1 for full GPS location data. Inset —The location of Utila is indicated with a black circle relative to the western Caribbean and Gulf of Mexico.

in the PERMANOVA (*Anderson, Gorley & Clarke, 2008*). When processing samples, one 15 m Black Coral Wall light trap collected no taxonomic groups with sufficient biomass to register on our field scales (weight < 0.01 g); this necessitated its removal from multivariate analysis of biomass data. All PERMANOVAs were run for 99999 permutations using the 'adonis' function in vegan (*Oksanen et al., 2015*) in R (*R Core Team, 2013*). Constrained analysis of principal coordinates (CAP) was conducted for the abundance data using the 'capscale' function in vegan (*Oksanen et al., 2015*). All taxonomic groups with a Pearson correlation coefficient |< 0.5| with either of the first two CAP axes were identified as potential drivers of community difference with depth. The abundance of these taxonomic groups was then individually tested using a Euclidian PERMANOVA to identify whether they changed with depth.

## RESULTS

Overall we found similar richness of taxonomic groups on shallow and mesophotic reefs (Fig. 2A), with much variation in the overall abundance and biomass at both depths (Figs. 2B and 2C ). We used an NMDS to visualise differences in the community sampled by the light traps at shallow and mesophotic depths. Abundance data appeared to show

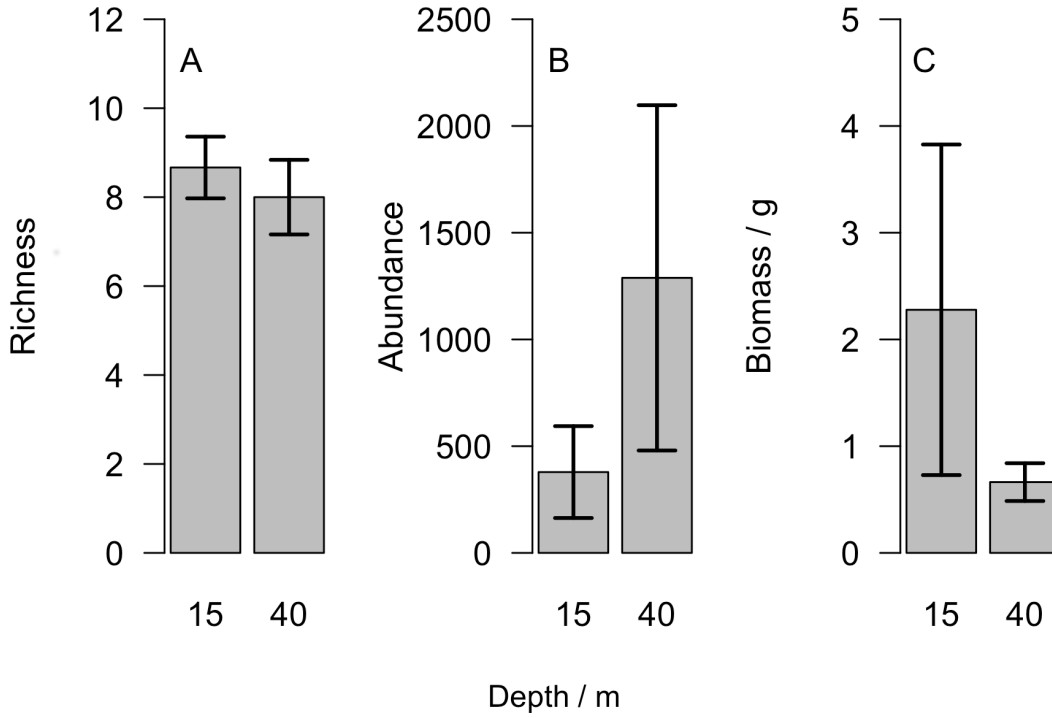

**Figure 2** (A) Number of different taxonomic groups recorded, (B) mean abundance per light trap and (C) total biomass across all taxa groups comparing reefs at 15 m and 40 m. Bars show mean ±1 standard error.

a difference in the community between shallow and mesophotic reefs (Fig. 3A), while biomass-weighted data showed no clear patterns (Fig. 3B). We tested these patterns in a PERMANOVA (Table 1), finding differences in the recorded light trap community taxonomic richness, abundance and biomass between survey sites. Abundance of taxonomic groups also changed between the two depths, but no pattern with depth was found for taxonomic richness or biomass.

We conducted Euclidian PERMANOVAs on abundance results correlating with the CAP axis to identify taxonomic groups varying with depth (Table 2). We identified decapod crab zoea, mysid shrimps, peracarid crustaceans and oligochaete abundance as increasing on MCEs compared to shallow reefs, with no oligochaetes recorded on shallow reefs. We did not detect any zooplanktivorous groups at greater abundance on shallow reefs than MCEs, nor any changes in fish larvae abundance between shallow reefs and MCEs.

## DISCUSSION

While MCEs are of increased interest because of their potential role as refuges for threatened shallow-reef taxa (*Bongaerts et al., 2010*; *Bridge et al., 2013*; *Lindfield et al., 2016*), almost all existing research has focused on large sessile benthic taxa such as hard corals, macroalgae and sponges, or large mobile organisms such as fishes (*Kahng et al., 2010*; *Kahng, Copus & Wagner, 2014*). Few studies have considered changes in small mobile invertebrates making up reef cryptofauna and zooplankton. We found significant differences in zooplankton

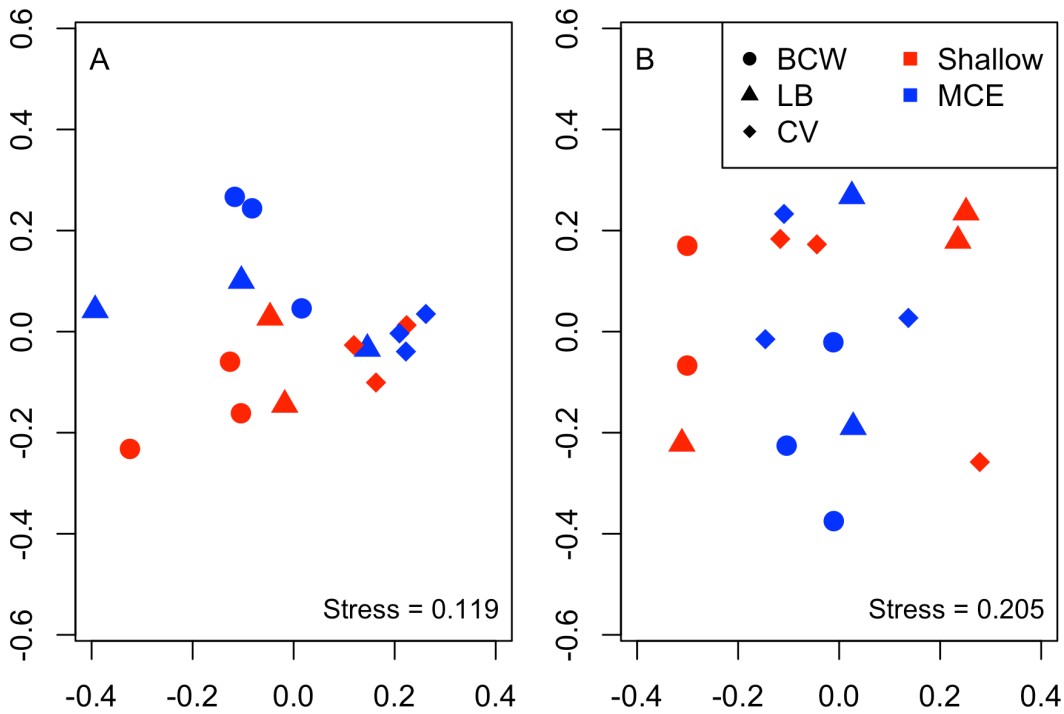

**Figure 3** **Nonmetric multidimensional scaling plot for (A) abundance and (B) biomass of the invertebrate and fish larvae.** Colours indicate different depths, while shapes indicate different survey sites. Sites were: BCW, Black Coral Wall; LB, Little Bight and CV, Coral View.

**Table 1** **PERMANOVA results testing for differences in the recorded light trap community between sites and depth for abundance and biomass data.**

|  | DF | Richness | | Abundance | | Biomass | |
|---|---|---|---|---|---|---|---|
|  |  | pseudo-F | P | pseudo-F | P | pseudo-F | P |
| Survey site | 2 | 3.97 | 0.04 | 5.58 | <0.0001 | 2.33 | 0.01 |
| Depth | 1 | 0.76 | 0.41 | 2.71 | 0.03 | 1.42 | 0.24 |
| Residual | 13 |  |  |  |  |  |  |
| Total | 16 |  |  |  |  |  |  |

richness, abundance and biomass between study sites and, interestingly, higher abundances on MCEs than on shallow reefs while biomass did not change.

Previous work has suggested zooplankton, in particular larger planktonic individuals such as mysids, isopods and decapod larvae are particularly important prey items for planktivorous fish (*Hobson & Chess, 1978*). Unlike in many locations (see *Kahng et al., 2010*; *Kahng, Copus & Wagner, 2014*), a previous study on Utila at our study sites found no difference in relative planktivorous reef fish biomass between shallow reefs and MCEs, and a decline in actual biomass of planktivorous reef fish with increased depth (*Andradi-Brown et al., 2016b*). This pattern is surprising as we identified greater abundance of zooplankton >2 mm size and no change in biomass of zooplankton across the depth gradient, suggesting similar or potentially greater food resources for planktivorous fish on MCEs. In addition,

**Table 2  Euclidian PERMANOVA results for abundance of taxonomic groups that correlate |>0.5| with the first or second CAP analysis.**

| Taxonomic group | Shallow (15 m) | | Mesophotic (40 m) | | *Pseudo-F* | *P* |
|---|---|---|---|---|---|---|
| | **Mean** | **SE** | **Mean** | **SE** | | |
| *First axis* | | | | | | |
| Decapod crab megalopae | 25.00 | 18.92 | 15.78 | 14.13 | 0.47 | 0.24 |
| Decapod crab zoea | 17.89 | 17.39 | 69.11 | 56.53 | 2.31 | <0.01 |
| Decapod shrimp zoea | 52.00 | 19.66 | 38.22 | 26.72 | 0.27 | 0.53 |
| Mysid shrimps | 84.11 | 63.12 | 637.44 | 600.61 | 1.52 | 0.04 |
| Peracarid crustaceans | 68.22 | 44.79 | 216.11 | 93.51 | 5.04 | <0.01 |
| Urochordates | 29.11 | 29.11 | 83.89 | 74.11 | 1.32 | 0.15 |
| *Second axis* | | | | | | |
| Cladocerans | 4.33 | 4.33 | 0.33 | 0.33 | 2.44 | 0.10 |
| Oligochaetes | 0.00 | 0.00 | 93.67 | 93.67 | 2.28 | <0.01 |

planktivorous reef fish exhibit high visual system plastic adaptive ability, and show few differences in feeding ability caused by changes in light levels across depth gradients, suggesting they should be able to feed efficiently at MCE depths included in this study (*Brokovich et al., 2010*). However, the previous planktivorous fish study from Utila was conducted during daylight hours (*Andradi-Brown et al., 2016b*), whilst light traps in this study were deployed overnight. Therefore, as zooplankton are known to have diurnal movement patterns (*Angel, 1985*), the abundance of zooplankton we recorded here may not be available to planktivorous fish during daylight hours. MCEs on the south shore of Utila exist as a gently sloping patch reef system on the continental shelf that remains within mesophotic depths before rising to become the mainland of Honduras. Therefore, unlike other MCEs adjacent to deep-sea habitats, at the sites we surveyed there is unlikely to be a large diurnal migration of zooplankton from deeper water at night.

Our finding of greater abundance of some zooplanktivorus groups on MCEs, and no change in zooplankton biomass between shallow reefs and MCEs contrast with previously identified zooplankton depth patterns. In Jamaica, *Ohlhorst (1985)* studied zooplankton across a 6–24 m depth gradient using traps placed over the reef, finding that both the abundance of zooplankton and the volume of zooplankton per trap declined with depth. When looking at specific taxonomic groups, we identified greater abundance of mysid shrimps and peracarid crustaceans on MCEs (40 m) than shallow reefs (15 m). Whereas, no difference was identified in mysid shrimp or peracarid crustacean abundance between 15 m and 24 m in Jamaica (*Ohlhorst, 1985*). In Hawaiian reef cryptofauna, brachyuran crab abundance has been reported to decline across a 12–90 m depth gradient (*Hurley et al., 2016*). However much of this pattern was caused by one crab genus, which when excluded led to crab abundance increasing with depth. While these Hawaiian brachyuran crabs had settled on the reef, we identified increased abundance of decapod crab zoea on MCEs on Utila, though no difference in decapod crab megalopae with depth. In addition, we found similar abundances of fish larvae between shallow reefs and MCEs. However, fish larval recruitment is

known to be seasonal, and previous studies have identified abundance differences in fish larval recruits across 10–40 m in the Caribbean (*Luckhurst & Luckhurst, 1977*). These fish recruitment patterns were highly species specific, with fish recruits more abundant for some species at 40 m than 10 m, whilst the reverse is true for other species (*Luckhurst & Luckhurst, 1977*).

In this study we found no change in taxa richness between shallow reefs and MCEs, however, we only classified invertebrates into broad taxonomic groups, lacking the resolution needed to detect fine scale richness patterns. In Jamaica, *Ohlhorst (1985)* reported an increase in taxonomic richness across depths from 6 to 24 m but used higher resolution taxonomic groupings. In contrast, in Hawaii, *Hurley et al. (2016)* reported the greatest brachyuran crab reef cryptofauna diversity on shallow reefs, with 40% of species at 12 m and declining richness with depth to 90 m. Differences in richness patterns with depth between these studies are likely caused by different reef habitats, taxonomic resolution, biogeographic regions, and sampling techniques

Patterns in species richness across the shallow to mesophotic gradient has been a major focus of research (*Kahng et al., 2010*; *Kahng, Copus & Wagner, 2014*), potentially being used to inform conservation management and in defining MCE ecology (*Laverick et al., 2016*). The current upper depth limit of MCEs is defined at 30–40 m based on the limits of recreational SCUBA diving (*Menza, Kendall & Hile, 2008*; *Hinderstein et al., 2010*; *Loya et al., 2016*). However, there is disagreement within the mesophotic research community over whether this upper limit should be redefined based on biological community turnover with depth (*Laverick et al., 2016*). Recently a consensus formed that MCEs can be divided into upper and lower zones based on a species transition commonly observed in scleratinian corals and fishes at approximately 60 m depth in many locations (*Loya et al., 2016*). At present, with MCE biodiversity so poorly documented, including the near absence of studies on zooplankton and mobile reef invertebrates, further investigative work is necessary to test whether the recorded patterns in reef communities with depth are consistent across many other taxonomic groups on reefs

This study provides a first glimpse of the patterns in zooplankton communities associated with MCEs in the Caribbean. Further research is necessary to determine fine-scale patterns across the depth gradient in zooplankton communities to help identify depth transition zones between communities and areas with unique biodiversity assemblages.

## ACKNOWLEDGEMENTS

We thank J Hogg and A Price at the Department of Zoology, University of Oxford workshop for light trap construction. We thank all the staff and students of Operation Wallacea Utila Marine Program 2015 for help and support with deploying the light traps.

### Funding

DAAB is funded by a Fisheries Society of the British Isles PhD studentship (http://www.fsbi.org.uk). Operation Wallacea (http://www.opwall.com) provided fieldwork

support for DAAB, DAE, CLH, AH, EG and ADR. Operation Wallacea provided financial support in the form of salaries for authors DAAB, DAE, AH and EG. The funders had no role in study design, data collection and analysis, decision to publish, or preparation of the manuscript.

### Grant Disclosures

The following grant information was disclosed by the authors:
Fisheries Society of the British Isles PhD studentship.
Operation Wallacea.

### Competing Interests

Operation Wallacea provided financial support in the form of salaries for authors DAAB, DAE, AH and EG, but did not have any additional role in the study design, data collection and analysis, decision to publish, or preparation of the manuscript.

### Author Contributions

- Dominic A. Andradi-Brown conceived and designed the experiments, performed the experiments, analyzed the data, contributed reagents/materials/analysis tools, wrote the paper, prepared figures and/or tables.
- Catherine E.I. Head reviewed drafts of the paper.
- Dan A. Exton and Alex D. Rogers conceived and designed the experiments, contributed reagents/materials/analysis tools, reviewed drafts of the paper.
- Christina L. Hunt, Alicia Hendrix and Erika Gress performed the experiments, reviewed drafts of the paper.

### Animal Ethics

The following information was supplied relating to ethical approvals (i.e., approving body and any reference numbers):

This study was on zooplankton communities sampled on coral reefs in Honduras. This meant the primary study organisms were invertebrates, though by the nature of the sampling, fish larvae were also collected. As the focus of the work was on invertebrates, and no higher vertebrates were involved, ethical review was not required for this work in Honduras.

### Field Study Permissions

The following information was supplied relating to field study approvals (i.e., approving body and any reference numbers):

Research permits for this work were issued to Operation Wallacea by the Instituto de Conservación Forestal (ICF), Honduras, permit number: ICF-261-16.

### Data Availability

The raw data has been supplied as a Supplemental File.

## Supplemental Information

Supplemental information for this article can be found online at http://dx.doi.org/10.7717/peerj.2853#supplemental-information.

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
