# Peer review of "Identifying zooplankton community changes between shallow and upper-mesophotic reefs on the Mesoamerican Barrier Reef, Caribbean"

_PeerJ, doi:10.7717/peerj.2853_

## Round 0.1 · original submission · Minor Revisions

Please consider all the suggestions in the revised manuscript version. Furthermore, the English should also be checked.

·

Basic reporting

Overall, this manuscript is well-written and adheres closely to the templates and instructions for authors of PeerJ. The writing is clear and succinct, and needs few edits. My primary comments surround additional literature citations that would add some depth and breadth to the introduction and discussion sections.

Line 27: The authors cite Hinderstein et al. 2010. This was not a review paper, it was the equivalent of an introductory chapter to a series of workshop-produced papers. I suggest citing the primary literature here to support the statement on the depth range of light-dependent corals, e.g. Maragos and Jokiel 1986, Kahng and Maragos 2006.

Line 57: The authors should start with a geographic review that is greater than just the Caribbean. Consider citing Brockovich et al. 2008 and similar papers first to illustrate that dominance of plankivores in MCE fish assemblages is a global phenomenon.

Line 65: In addition to Lesser et al. 2010, I suggest citing other earlier papers such as Mass et al. 2007 and Fricke et al. 1987 to support the possibility of increasing heterotrophy with depth.

Line 153: I suggest citing additional references on the MCE refugium hypothesis, e.g. Hoegh-Guldberg et al. 2007, Lindfield et al. 2016.

Experimental design

Overall, the field sampling techniques were appropriate to the question at hand. As the authors note, however, this method only sampled nocturnal plankton biomass and diversity. Since this was stated up front, the results are what they are, and can stand on their own. I do not see this as a shortcoming of the study.

40 m depth is barely mesophotic, and some disagreement exists within the mesophotic research community regarding whether a one-size-fits-all upper depth limit is appropriate (especially given that some systems seem to show a pronounced turnover in species composition at 50+ m). This fact could be acknowledged somewhere in this paper. Nevertheless, the findings here indicate that there are differences between 15 m and 40 m plankton communities, and these results can stand on their own without quibbling over the definition of mesophotic.

Line 92: When two traps were deployed at a given site on a single night, how far apart were they? How far away is the light emitted by the LEDs visible underwater?

Line 101: It would be useful to note whether samples were sorted using the naked eye, or using some sort of optical enhancement such as a dissecting scope.

Validity of the findings

The statistical analyses are robust, and are appropriate to the question and data sets at hand. Presenting the data as both bar charts and nMDS plots give the reader a well-rounded view of how the 15 and 40 m communities differ.

Additional comments

Most (if not all) mesophotic papers begin with a statement regarding the scarcity of data from MCEs, and how a large majority of what is known about coral reefs is derived from the upper 20% of their depth range. The authors take this one step further, and point out that within the mesophotic research field, there are numerous undercharacterized communities and taxa. The structure of zooplankton communities is one of the biggest gaps in knowledge of MCEs, and yet I would guess that most mesophotic researchers have considered plankton in only the most cursory of manners. Understanding the structure of the zooplankton communites across a depth range is critical to our understanding of major ecological themes, including the role of heterotrophy for hermatypic corals at depth, and the dominance of planktivorous fishes at mesophotic depths.

Thus, although this paper poses a very simple, straightforward question, I believe its importance to the MCE field far outweighs the manuscript’s modest length. I congratulate the authors on a simple, well-done study that will constitute a significant contribution to the field.

·

Basic reporting

- Use "fishes" as plural when referring to multiple species (e.g., abstract "...or mobile taxa such as fishes.")

- There are some slightly inappropriate uses (or non-uses) of commas and other sentence structure and grammar (e.g., "coral reef" when used as a compound adjective needs a hyphen, but when used as a noun does not), but not significant enough to warrant individual corrections. It might be helpful if the authors sought the review of an experienced editor to ensure correct punctuation and grammar throughout.

Experimental design

No comments

Validity of the findings

The results are interesting and appropriately analyzed. They help address largely unknown questions about zooplankton distribution across both shallow and MCE reefs, and represent an interesting contrast to the few other studies on this issue in other regions.

Additional comments

No Comments

---

## Round 0.2 · accepted · Accept

Thank you for improving your manuscript and publishing in PeerJ.